# Study protocol of a breathing and relaxation intervention included in antenatal education: A randomised controlled trial (BreLax study)

Vanessa Leutenegger[1,2]*, Frank Wieber[3,4©], Deirdre Daly[5‡], Leila Sultan-Beyer[6‡], Jennifer Bagehorn[7‡], Jessica Pehlke-Milde[1©]

**1** School of Health Sciences, Institute of Midwifery and Reproductive Health, ZHAW Zurich University of Applied Sciences, Winterthur, Switzerland, **2** Faculty of Medicine, University of Zurich, Zurich, Switzerland, **3** School of Health Sciences, Research Institute of Public Health, ZHAW Zurich University of Applied Sciences, Winterthur, Switzerland, **4** Department of Psychology, University of Konstanz, Konstanz, Germany, **5** School of Nursing and Midwifery, Trinity College Dublin, Dublin, Ireland, **6** KSW Cantonal Hospital Winterthur, Winterthur, Switzerland, **7** School of Management and Law, Centre of Customer Experience & Service Design, ZHAW Zurich University of Applied Sciences, Winterthur, Switzerland

© These authors contributed equally to this work.
‡ DD, LSB and JB also contributed equally to this work.
* vanessa.leutenegger@zhaw.ch

**Data Availability Statement:** No datasets were generated or analysed during the current study. All

## Abstract

### Background

Antenatal education is part of antenatal care in many countries. Physical and mental preparation for childbirth and information on pain management are considered key elements of antenatal education classes. Evidence suggests that women who participate in antenatal education classes appear to benefit in terms of self-efficacy and childbirth experience. In particular, women with increased fear of childbirth benefit from trained breathing and relaxation techniques in antenatal education. However, little is known about the effect of breathing and relaxation techniques on the outcomes of healthy pregnant women without fear of childbirth or who do not have medical or obstetric risks, or on neonatal outcomes.

### Objective

The aim of this study is to test whether including a breathing and relaxation technique in an antenatal education class will improve self-efficacy towards birth compared to a standard antenatal education class.

### Methods

The study will be a two-armed randomised controlled trial (RCT). Healthy pregnant women between the 12th and 30th week of gestation with singleton low-risk pregnancies and who are receiving routine antenatal care will be recruited in a hospital in Switzerland. All women included will attend an 8-hour antenatal education class. The intervention group will additionally practise a breathing and relaxation technique, receive a handbook to guide their exercise practice at home, and be provided with access to an online brochure with video

relevant data from this study will be made available upon study completion. The decision on which repository the data will be made available is still pending.

**Funding:** The study is co-funded by the Amaari Foundation. The funder supported the author (VL) in the form of third-party funding but played no additional role in the design of the study, the data collection and analysis, the decision to publish or the preparation of the manuscript.

**Competing interests:** The authors have declared that no competing interests exist.

and audio recordings for guidance. Data on maternal and neonatal outcomes will be collected after recruitment, in the 37th week of pregnancy, and two to four weeks postpartum.

## Discussion

The effectiveness of including a breathing and relaxation technique in antenatal education classes on women's self-efficacy is discussed as a means to improving women's pregnancy and childbirth outcomes.

## Trial registration number

NCT06003946, SNCTP000005672.

## Introduction

Antenatal education classes were developed to inform expectant mothers about pregnancy, labour and birth, and the postpartum period, in order to improve the pregnancy and childbirth experience [1, 2]. Originally, the classes were based on the concepts of, among others, Lamaze [3] and Grantly Dick-Read [4, 5]. Studies indicate positive emotional effects on labour and birth outcomes in women who have attended antenatal education classes. These include lower levels of maternal stress, higher levels of self-efficacy, lower levels of caesarean birth rate and the use of epidural anaesthesia. However, the effectiveness of this method in improving maternal and neonatal outcomes is limited, as is what parts of antenatal education might have an effect [6].

In higher and middle-income countries, women and their partners are offered antenatal education classes as part of antenatal care. In most instances, the classes are offered by midwives and consist of a series of sessions. Additional courses may be offered by women's health physiotherapists and other women's health professionals, such as obstetricians. The classes are usually held as a weekly course, with four to ten evening courses or one to three weekends. However, the content of antenatal classes, as well as the focus and duration, varies widely. The main purpose of the classes is to increase knowledge and confidence in relation to pregnancy, labour, and birth, as well as the postpartum period [7], principally through the provision of information on and preparation for the management of labour pain, by means of non-pharmacological techniques such as breathing and relaxation techniques [8]. In particular, a key factor in achieving this aim is the promotion of self-efficacy so that women will feel confident that they are in control of labour and able to manage their labour pain [2]. For instance, Howarth and Swain [9] were able to show in their randomised controlled trial that women apply trained practical body skills such as breathing and relaxation techniques according to their individual needs, which consequently increases their ability to feel in control. Despite the important role self-efficacy has to play in women's ability to cope with labour and birth, it has received little attention in the development of antenatal education [2]. It is therefore critical that antenatal education be systematically developed to incorporate breathing and relaxation techniques to improve self-efficacy towards birth.

In maternity care, childbirth preparation practices are based on experience but have rarely been systematically developed, implemented, and evaluated [10]. This, in part, explains the heterogeneity of the results of their effectiveness. Although several studies have shown positive effects on women's self-efficacy, including lower rates of epidural anaesthesia use and recall of labour pain [9, 11–16], the overall evidence for a positive association between antenatal preparation and neonatal outcomes remains equivocal [8, 17–19].

At present, the best available evidence on effects of antenatal education classes focussing on elements such as breathing and relaxation techniques is from research with women who fear childbirth or who are experiencing mental health issues. For these women, antenatal classes strengthened their resources, had a positive effect on pregnancy outcomes and their childbirth experience, and enabled them to be competent and proactive during childbirth [20, 21]. Frequent practise of breathing and relaxation techniques led to women diagnosed with mental health issues such as stress and anxiety feeling better able to manage labour pain and increase their levels of self-efficacy [22].

Given that antenatal education classes are able to have a positive effect on levels of self-efficacy, maternal stress, lower rates of epidural anaesthesia, and reduce recall of labour pain in women without fear of childbirth, we propose the development of an education module focusing on breathing and relaxation technique for inclusion in an antenatal education class. In addition, we propose that this class be assessed for impact on the levels of self-efficacy towards birth and other maternal and neonatal outcomes.

## Objective

The present study aims to test the effects of antenatal education inclusive of a specific training a breathing and relaxation technique (BreLax) on self-efficacy towards birth. Assessment will take place before and after the intervention and compared to the effects of a standard antenatal education class. The secondary objectives are as follows.

1. To identify the impact of BreLax on additional maternal outcomes before and after birth: women's satisfaction with their experience of childbirth, self-control, pain management, birthing position, duration of labour, and skin-to-skin contact > 1 hour.

2. To identify the effects of BreLax on neonatal outcomes: 5-minute Apgar-Score and umbilical cord pH.

3. To explore women's perceptions of the applicability of the intervention during labour.

## Methods

### Design

The study will be conducted as a population-based randomised controlled trial (RCT) with a pre-post design and two parallel arms. The control arm will consist of a standard antenatal education class (standard care) that contains information on the value of breathing and relaxation techniques and a few relaxation exercises but no training in a breathing and relaxation technique. The intervention arm will consist of a standard antenatal education class plus special training in a breathing and relaxation technique in class and access to an online manual for independent practice at home with audio and video files (Brelax).

Eligible participants will be randomly selected to complete either the control or intervention arm. Three repeated measurements will be performed and, in addition, labour and birth data will be transferred from the documentation system: before intervention (T0); after intervention (T1); labour and birth data outside the documentation system (T2) and two to four weeks after birth (T3) (see Fig 1, S1 Checklist).

### Setting

This research will be carried out in a Swiss regional hospital with an annual birth rate of around 1800. Antenatal education classes are offered as part of the hospital's facilities.

| | Study Period | | | | |
|---|---|---|---|---|---|
| | Enrolment | Allocation | Post-allocation | | |
| **TIMEPOINT** | *-t₁* | **0** | *t₁* | *t₂* | *t₃* |
| **ENROLMENT:** | | | | | |
| **Eligibility screen, FOBS scale** | X | | | | |
| **Informed consent** | X | | | | |
| **Allocation** | | X | | | |
| **INTERVENTIONS:** | | | | | |
| *Intervention group* | | X | X | | |
| *Control group* | | X | X | | |
| **ASSESSMENTS:** | | | | | |
| Age, nationality, living situation, education, employment, family income, gestational age, parity, obstetric history, medical history | | X | | | |
| *Primary outcome:* | | | | | |
| *self-efficacy* | | X | X | | |
| *Secondary outcomes:* | | | | | |
| *childbirth experience* | | | X | | X |
| *pain experience* | | | | | X |
| *labour and birth outcomes* | | | | X | |
| *birth preparation* | | | X | | X |

**Fig 1. SPIRIT schedule for enrollment, allocation, and post-allocation.**

## Participants

Pregnant women (age 18 or above) from 10 to 30 weeks of gestation, with a singleton low-risk pregnancy and receiving antenatal care. Exclusion criteria are 1) unwilling to attend an antenatal education class, 2) planning a caesarean section, 3) showing increased levels of fear related to childbirth measured by the fear of birth scale a two-item visual analogue scale (FOBS cut-off above 50) [23] and 3) not able to understand and speak German. Participants will be recruited during consultations of the hospital and clinic midwifes, in surrounding gynaecological as well as midwife practices, and through common social media channels. A website was created for recruitment, which is linked to the advertised antenatal courses on the clinic website. Interested women can register directly for further information. Women who have already registered for antenatal education classes will be informed directly by email.

A screening interview will be conducted with potential participants by telephone to verify eligibility, explain the study in detail, and address any concerns. Eligible participants will sign the informed consent form prior to the first antenatal education class.

## Sample size calculation

The sample-size calculation is based on the randomised controlled trial of a self-efficacy enhancing educational programme (SEEEP) by Ip et al., (2009) [2]. This study reported an effect size of 0.78 in increasing self-efficacy in labour and an attrition rate of around 30.7%. In the proposed study, we therefore assume an overall moderate effect size of 0.75. Setting a power at 80%, a significance level of $p < .05$, a sample size of 29 per arm is needed for a two-arm repeated measures design (GPower, version 3.1). Assuming an attrition rate of 20%, the final sample size will be 35 in each arm and 70 participants in total.

## Randomisation and allocation

Participants will be recruited in their first, second or early third trimester of pregnancy. After women have registered for an antenatal education class, their eligibility will be checked and

participants are enrolled. Due to the method of organisation of the course at the birth clinic, randomisation is conducted at the course level. Randomisation is carried out by the study midwife using envelopes containing the corresponding information about the course (intervention yes or no).

## Blinding

Participants will not be informed in which class they will be randomly placed and are accordingly unaware of the intervention, see Fig 2. However, complete blinding of behavioural interventions is difficult to guarantee as participants from both groups might meet outside the courses, talk about their classes and become aware of the differences. Blinding of midwives conducting antenatal classes is not possible. The collected data will be entered by a research assistant to ensure that data entry and analysis are not undertaken by the same people. The personal data of the participants will not be included in the data set [24].

## BreLax intervention

The intervention was designed in line with the Medical Research Council Framework (MRC) recommended four stages of research for the development and evaluation of complex health interventions [25] and using behaviour change techniques [26].

### Theoretical framework

**Self-efficacy theory.** The developed complex intervention is based on the theoretical concept of Bandura's self-efficacy theory [27], see Fig 3. According to Bandrua [27], self-efficacy is considered to be the expectation of whether an individual is capable of performing a particular behaviour in a given situation, which can be divided into outcome expectation and self-efficacy expectation. Outcome expectations refer to the individual's prediction of the possible outcomes of a set of behaviours. If someone predicts that a certain behaviour will lead to a certain outcome, that behaviour may be activated and chosen. Self-efficacy is primarily influenced by four sources of information, namely performance, vicarious experience, verbal persuasion, and physiological and emotional states [27].

According to Bandura [27] a strong belief in one's own ability to exercise some control over one's physical condition can serve as a psychological prognostic indicator of the likely level of health functioning [2]. Consistent with these assumptions, self-efficacy has been shown to have a significant influence on how labour is perceived and how it is physically managed [28]. Therefore, we assume that a strategy to cope with labour pain and increase self-control during birth can increase self-efficacy.

To increase women's self-efficacy towards birth and confidence in their ability to manage birth, the BreLax intervention is predicted to contribute to all four ways of improving self-efficacy. Women will likely have the experience of successfully performing breathing and relaxation techniques and see that other women can perform the techniques, too. They will also receive reassuring comments from class instructors and are likely to feel fewer distracting emotions and physical arousal. Using the online brochure, women will be motivated to continue practising at home. They will be guided through the exercises by audio and video instructions and receive reminders to keep practising.

**Behaviour change technique.** The intervention has been developed with a focus on behaviour change techniques and using the taxonomy of behaviour change (see S1 Table), as improving the implementation of evidence-based practices depends on behaviour change [26]. Therefore, behaviour change interventions are fundamental to effective practice. Behaviour change interventions can be defined as coordinated packages of measures aimed at changing

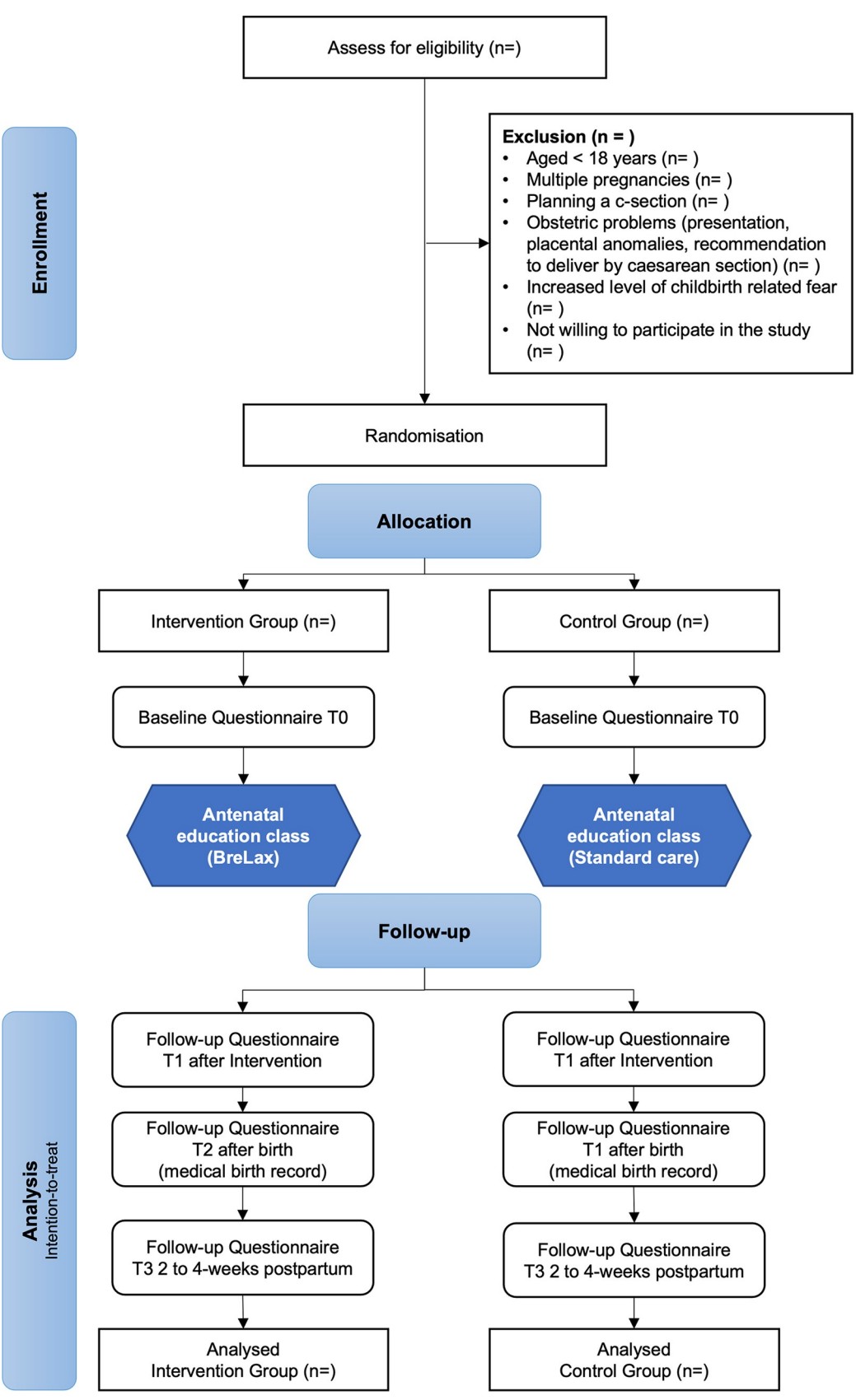

**Fig 2. Flowchart BreLax study.**

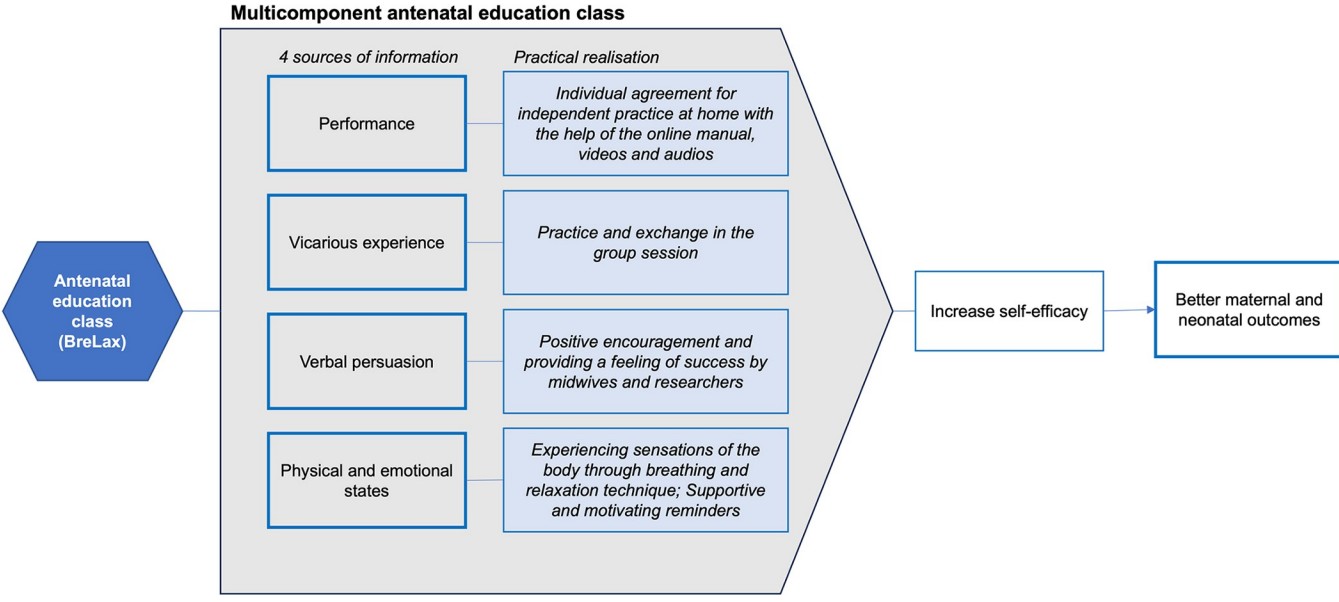

**Fig 3. How BreLax is supposed to improve self-efficacy.**

specific patterns of behaviour. For this purpose, the COM-B framework (capability, opportunity, motivation) has been developed as a supporting factor in the present study. Capability is defined as the psychological and physical ability of the individual to perform the activity in question [29]. In this intervention, this is achieved by practising and enabling exercises in the antenatal education class and by reminding participants to undertake independent practice at home. Opportunity is defined as all factors that lie outside the individual and enable or trigger the behaviour [29]. With the help of the atmosphere and the environment in the antenatal class, women are allowed to implement the behaviour, and through independent practice at home, awareness is created of what women individually need. Motivation is defined as all brain processes that stimulate and control behaviour, not just goals and conscious decisions [29]. Due to the imminent birth of their babies, women in antenatal education classes are motivated to deal with antenatal topics and motivated to strengthen their own abilities (see S1 Table).

**Antenatal education classes.** It is recommended that women start attending an antenatal education class in the second trimester or early in the third trimester. This recommendation applies to both the intervention and the control group. Participants will be randomly assigned to one of the two groups. One group serves as the control group and will receive a standard antenatal education class without the breathing and relaxation technique, while the intervention group will receive an antenatal education class inclusive of the designed BreLax intervention and the online brochure for practicing at home. Both week-long and weekend courses are offered. Each course has a duration of approximately eight hours spread over four weeks or a weekend. There will be six to twelve women in each class.

**Midwifery training.** The antenatal midwives in the birth clinic will be trained before the intervention. The investigator will organise a two-hour workshop in which the course concept and the breathing and relaxation technique will be discussed and practised together. In addition, the investigator will always be available to answer questions from the midwives. Midwives will receive a handout with study information and a description of the development of the breathing and relaxation technique and will be made aware of the importance of adhering to the procedures learnt in the workshop.

**Table 1. Content of antenatal education classes.**

| Content | BreLax-Model (intervention) | Standard care (control) |
|---|---|---|
| **Pregnancy** | | |
| Being pregnant / Doing pregnancy | ✓ | ✓ |
| **Labour and birth** | | |
| Labour and birth | ✓ | ✓ |
| Complications | ✓ | ✓ |
| Non-pharmalogical medications | ✓ | ✓ |
| Pharmalogical medications | ✓ | ✓ |
| Information about breathing and relaxation | ✓ | ✓ |
| BreLax intervention | ✓ | ✖ |
| BreLax online Brochure for home practice | ✓ | ✖ |
| Partner support | ✓ | ✓ |
| **Postpartum** | | |
| Baby care | ✓ | ✓ |
| Attachment | ✓ | ✓ |
| New role as parents | ✓ | ✓ |

**Informational component.** The intervention and standard care classes both aim to inform women about pregnancy, labour, birth, and the postpartum period. The main difference is that the intervention group will focus on breathing and relaxation technique (see Table 1). Midwives will instruct participants on how to do the exercises, how to perform the individual breathing pattern, and how to assume the respective positions and movements (S1 Fig). Additionally, women will receive a manual with exercises to practise at home two to three times a week for about five to ten minutes each. The control group will receive standard care about breathing and relaxation, but no joint exercise sequences in classes.

**Component of breathing and relaxation.** Breath awareness provides physical, mental, and emotional control. Deep breathing increases blood circulation, oxygen flow, and reduces stress, which is beneficial for both the mother and the baby. Through the learning of conscious breathing and relaxation techniques, women can more effectively control their pain and relax when uterus contractions begin, increasing confidence [30].

The core of the breathing technique is prolonged exhalation [31]. Such techniques have been taught in antenatal education for many years and are actively supported by midwives during labour and birth. In the present study, the focus is on prolonged exhalation with individual rhythm, which women will learn and practise during antenatal education.

The main advantage of this technique is that it helps in preparation for labour and birth, it can be actively used even in stressful situations, allows for the learning of an individual breathing pattern, and helps women to relax comfortably. Breathing technique refers to breathing with a certain number of repetitions and amplitudes [31]. Building on these techniques, women will be encouraged to continue practising at home using the online brochure for guidance.

Women need to learn how to adapt their breathing pattern to their individual needs. Breathing and relaxation techniques are only useful and practical if women can try them out and apply them in a range of everyday situations. Breathing and relaxation techniques can be used on any day and in any stressful or uncomfortable situation [32]. For this reason, it is important that women continue to practise at home. In the best case, breathing and relaxation techniques become routine or become so automated that the techniques are automatically remembered during birth and can be used accordingly. For a habit to form, it is necessary for

women to practise over a period of time [33]. Previous literature shows that an average of 66 days is required to recall similar automated exercises [34]. This time period coincides optimally with the recommended start of antenatal education classes in the second trimester.

**BreLax exercises.** The antenatal education class offers 30 to 45 minutes of joint practice. In addition, the women have access to audio and video instructions in the online brochure, which last between three and ten minutes. No special materials are required for the exercises. The exercises can be practised both at home and in different daily situations.

- Exercises to learn prolonged exhalation (following the 3–6 breathing technique for relaxation)

- Exercises for mental and physical relaxation (4 upright positions, standing, sitting, 4-foot, elevated lateral position, see S1 File)

- Optional: visualisation, music

## Scales and outcome measures

The primary outcome of the trial is self-efficacy. Further relevant maternal and neonatal outcomes will be summarised in Table 2, including the time points at which the measures will be taken. All questionnaire links will be sent to participants via email at each time point for them to fill out on their own devices.

## Statistical analysis

Quantitative data including scale measures will be collected at three timepoints as articulated above (See Table 2).

Data will be analysed using the intention-to-treat principle (ITT). If the missing data is >5%, the possibility of multiple imputation will be examined. We intend to minimise missing data and will, for example, design mandatory questions in the questionnaires. In addition, participants will be sent reminders to complete the follow-up questionnaires after two to four weeks.

**Table 2. Measures.**

| Variables | Measures | Timepoints |
|---|---|---|
| **Primary outcome** | | |
| Self-efficacy | Childbirth Self-Efficacy Inventory (CBSEI-32): The Childbirth Self-Efficacy Inventory is a questionnaire on self-efficacy expectations during childbirth and is based on the theory of Albert Bandura [35]. | T0, T1 |
| **Secondary outcomes** | | |
| Childbirth experience | Childbirth Experience Questionnaire (CEQ 2): The Childbirth Experience Questionnaire is used to measure the childbirth experience. The instrument measures the four dimensions of own capacity, perceived safety, professional support, and participation with a total of 22 items [36]. | T1, T3 |
| Pain experience | Visual analogue scales: VAS is considered a valid tool for assessing pain experienced during childbirth. The women self-reported the level of pain experienced during the three stages of labour using the range from 0, no pain at all, to 10, unimaginable pain. | T3 |
| Labour and Birth Results | Maternal: Gestational age at birth, pregnancy complications, mode of birth, pain management, self-control, mobility during labour, support during pregnancy and birth, skin-to-skin for > 1 hour<br>Neonatal: Birth weight, Apgar-score, umbilical cord pH | T2 |
| Birth preparation | Experience with antenatal education and home exercises, feasibility and practicability of breathing and relaxation technique through the online manual | T1, T3 |
| Sociodemographic | Age, nationality, living situation, education, employment, family income, gestational age, parity, obstetric history, medical history, birth preparation | T0 |

Sociodemographic and obstetric history will be presented using descriptive statistics including frequency, percentage, mean, standard deviation, median and percentiles. For normally distributed continuous variables, mean and standard deviation will be used as measures of central tendency and dispersion. For non-normal data the median will be used. Categorical data will be presented as frequency and percentage.

However, to examine the effect within each group, a serial trend analysis, such as a repeated measures ANOVA, will be performed from T0 to T3 for the primary and secondary outcome variables unless the residuals (errors) deviate significantly from the normal distribution. If the data are non-normally distributed, alternative tests such as the Friedman test will be used to analyse the interaction of groups over time.

A series of linear mixed-effects models (LMMs) will be used to assess mediation effects. If the assumptions for using LMMs are not met, non-linear mixed-effects models or, in the case of non-normally distributed data, generalised linear mixed models will be used.

Quantitative data are analysed using R. The significant level is set at $p < 0.05$.

## Data monitoring

Given the timeframe and the minimal risk known to the participants in the current study, a data monitoring committee was not formed. The research team will continue to assess the need for the formation of such a committee.

## Data management

The investigator and the research assistant will be responsible for data collection and the data will be entered directly into a secure web-based database developed to support data collection for Research Studies with Built-In Domain Controls (REDCap) (project-redcap.org). The access of the user to the database is restricted and assigned by the investigator. Data will be entered into the database with a unique trial number and no identifiable data will be stored in the database. Data with invalid trial numbers, out-of-range values, or follow-up IDs that do not match the baseline trial number will be excluded.

## Ethics

This study has been approved by the Ethics Committee of Zurich (SNCTP000005672). Participants in both the intervention and control groups will participate voluntarily in the present study. Participants in the control group will receive the standard antenatal education class (standard care). In both the intervention and control groups, participants will receive written informed consent after being informed. Participants in both groups will complete online questionnaires listed in the "Measures" section of the article. Any changes to the protocol are subject to formal amendment and may not be implemented prior to approval by the Ethics Committee of Zurich.

## Risk assessment

We do not expect any serious risks to participants. Antenatal education is part of antenatal care and is financed by obligatory health insurance (OKP) in Switzerland. Every woman has the opportunity to attend an antenatal education class if she wants to. Women benefit from the information provided in the antenatal class, as well as the exercises to support them during labour and birth. However, participants will be required to complete questionnaires during pregnancy and after birth. This could be an additional challenge and a higher workload for women. Additionally, it is possible that the negative experience of childbirth might return

when completing questionnaires. Therefore, participants are offered the opportunity to contact the supervisor midwife and have a follow-up conversation with the attending birth midwife. If the need for further support is indicated, it is possible to involve the relevant specialists in the clinic.

## Discussion

The purpose of this study is to determine whether the inclusion of a breathing and relaxation technique in an antenatal education class can enhance self-efficacy towards birth, in comparison to a standard antenatal education class. To our knowledge, our study will be the first RCT to assess a multi-component intervention such as this.

The planned study aims to provide evidence regarding the potential benefits of antenatal education with a focus on breathing and relaxation techniques for pregnant women in general without increased fear of childbirth. It will provide evidence on whether antenatal education classes with a focus on breathing and relaxation techniques effectively prepare women for labour, birth, and pain management.

In addition to the results, the strengths and limitations of the study will be discussed. Strengths include the study design of a RCT to ensure sufficient power to demonstrate the effectiveness of the BreLax intervention. Furthermore, the study intervention will be systematically developed based on the concept of self-efficacy and the evidence on behaviour change techniques. The intervention will be taught and practiced together in direct face-to-face contact by the midwives and will be supported by an online brochure to help women to continue independent practice at home.

One challenge and potential limitation is that birth is an unpredictable and complex process, and unexpected complications may occur, which might affect the outcome for the mother and child, as well as influence the mother's childbirth experience. In addition, the quality of care the women will receive will be influenced by multiple factors such as respectful and effective communication and the support of qualified empathetic staff. This study is not designed to explain how these factors affect women's ability to cope with labour pain. Furthermore, additional research will be needed to understand moderating factors, such as partner support, training frequency and personal interaction between the midwives and participants.

The BreLax intervention has the potential to be a promising element of antenatal preparation, as the methodology promotes standardisation and reproducibility. It focuses on an essential component of antenatal preparation, namely breathing and relaxation techniques, and the effectiveness of this will be tested. Digitalisation within the healthcare sector will also be considered with the use of online handbook and reminders.

## Status and schedule of the study

The study began recruiting participants in December 2023, with enrolment planned over 6–10 months. The end of the study is defined when the postnatal questionnaire has been received from all participants, but no later than 4 months after inclusion of the last patient.

## Supporting information

**S1 Checklist. SPIRIT checklist BreLax.**
(PDF)

**S1 Table. Taxonomy of the behaviour change technique of the BreLax intervention.**
(PDF)

**S1 Fig. Upright positions BreLax.**
(TIF)

**S1 File. BreLax manual women (translated).**
(PDF)

**S1 Data.**
(PDF)

## Acknowledgments

The authors are grateful to the clinical site for their contributions to the implementation of the study.

## Author Contributions

**Conceptualization:** Vanessa Leutenegger, Frank Wieber, Jessica Pehlke-Milde.

**Funding acquisition:** Vanessa Leutenegger.

**Methodology:** Vanessa Leutenegger, Frank Wieber, Jessica Pehlke-Milde.

**Project administration:** Vanessa Leutenegger.

**Resources:** Vanessa Leutenegger.

**Writing – original draft:** Vanessa Leutenegger.

**Writing – review & editing:** Frank Wieber, Deirdre Daly, Leila Sultan-Beyer, Jennifer Bagehorn, Jessica Pehlke-Milde.

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
