## [Decision Letter · Decision Letter 0]

5 Jun 2024

PONE-D-24-06197Study protocol of a breathing and relaxation intervention in antenatal education: a randomised controlled trial (BreLax study)PLOS ONE

Dear Dr. Leutenegger,

Thank you for submitting your manuscript to PLOS ONE. After careful consideration, we feel that it has merit but does not fully meet PLOS ONE’s publication criteria as it currently stands. Therefore, we invite you to submit a revised version of the manuscript that addresses the points raised during the review process.

We look forward to receiving your revised manuscript.

Kind regards,

David Chibuike Ikwuka, Ph.D.

Academic Editor

PLOS ONE

3. We notice that your supplementary tables are included in the manuscript file. Please remove them and upload them with the file type 'Supporting Information'. Please ensure that each Supporting Information file has a legend listed in the manuscript after the references list.

Reviewers' comments:

Reviewer's Responses to Questions

**Comments to the Author**

1. Does the manuscript provide a valid rationale for the proposed study, with clearly identified and justified research questions?

Reviewer #1: Yes

Reviewer #2: Yes

2. Is the protocol technically sound and planned in a manner that will lead to a meaningful outcome and allow testing the stated hypotheses?

Reviewer #1: Partly

Reviewer #2: Yes

3. Is the methodology feasible and described in sufficient detail to allow the work to be replicable?

Reviewer #1: No

Reviewer #2: Yes

4. Have the authors described where all data underlying the findings will be made available when the study is complete?

Reviewer #1: No

Reviewer #2: No

5. Is the manuscript presented in an intelligible fashion and written in standard English?

Reviewer #1: Yes

Reviewer #2: Yes

6. Review Comments to the Author

You may also provide optional suggestions and comments to authors that they might find helpful in planning their study.

Reviewer #1: The paper is quite well written and well organized. I confine my comments to statistical matters.

L174: Why verify eligibility by phone? Are there no baseline (clinical) variables to be collected at the onset? Or values of baseline variables have to meet certain threshold before admission to the trial? If so, eligibility can be confirmed on site during the baseline interview.

L180: What is the interpretation of an effect size of 0.75 in the context of the trial.

L183: The 20% attrition rate comes from nowhere without justifications. This results in a sample size of of 35 subjects to be recruited for each arm. The hospital has 1800 births and there was a brief discussion of contamination. So why not perform the trial staggered over time to minimize contamination?

L189: Why have the midwife use envelopes to randomize subjects into group membership. It is the same reason why you don’t want the same statistician to randomize the trial and perform the analysis.

L334 ‘the possibility of multiple imputation’ will be examined. This is very vague. Please elaborate. Identify situations if and when you will use multiple imputation.

L344 You also reference doing the analyses using a serious of linear mixed effects models. What models do you have in mind? Please provide more details and display the regression models and show what are the random and fixed effects for added clarity

Further comments.

Throughout, the methodology seems to assume the outcome is normally distributed. Where are the checks and what is the proposed methodology if they are not normally distributed?

(I actually don't understand why we need to fill in this box when the full report is enclosed as a pdf).

Reviewer #2: The abstract by PLOS ONE's general guideline should not exceed 300 words; please summarise to reduce the words. Additionally, you used Harvard reference style; PLOS uses the Vancouver” reference style. Please, authors should take note of this and do the needful throughout the text and apply same to the list of references by numbering in the order in which they appear in the text. The reference numbers in the text should be in brackets. There were one or two spelling errors and missing word(s), e.g. "pharmalogical" (line 89) which I believe ought to be pharmacological. Authors should please read the manuscript thoroughly and religiously to correct the errors so observed. How do the authors reconcile the statement (lines 255 and 256) under antenatal education classes with that in randomisation and allocation (lines 186 and 187)?

7. PLOS authors have the option to publish the peer review history of their article (what does this mean?). If published, this will include your full peer review and any attached files.

Reviewer #1: No

Reviewer #2: **Yes: **Dr Stanley M Maduagwu

---

## [Author Response · Author response to Decision Letter 0]

5 Jul 2024

Responses to reviewers' comments (also shown in the "Response to reviewers" document) 

2. 1. Is the protocol technically sound and planned in a manner that will lead to a meaningful outcome and allow testing the stated hypotheses? 

Thank you for this advice. As we recognise that some aspects of the methodology and analysis may need to be refined as the work progresses, we will set out potential assumptions from the outset. This will provide transparency about the principles of our research approach. In addition, we will describe which aspects of the analyses we propose are exploratory. This distinction between confirmatory and exploratory analyses is important to ensure the integrity of our findings. 

For this reason, we have substantiated the data analysis.

3. Is the methodology feasible and described in sufficient detail to allow the work to be replicable?

Thank you for the useful advice, which we have implemented as follows in relation to the sample size calculation.

The sample size calculation was based on a previous study (Ip et al., 2009), and the effect size of 0.75 used for this was therefore a rather conservative figure. The attrition rate of 20% was based on the assumption that the women are closely monitored by the midwives and the specialists involved in the clinic, which is why it is assumed that pathological pregnancies are more likely to be the reason for possible exclusions. In addition, participation in the study is not very time-consuming with an antenatal education class as an intervention and three 20-minute questionnaires. 

Accordingly, L338 till L350 are defined in more detail so that repeatability and transparency are ensured.

4. Have the authors described where all data underlying the findings will be made available when the study is complete?

Thank you for pointing this out, this issue was clearly not addressed enough. No data sets were generated or analyzed during the current study. All relevant data from this study will be made available upon completion of the study. 

The ethics protocol was made available to the public on clinicaltrial.gov. 

Reviwer 1#: Further comments: 

L174: Why verify eligibility by phone? Are there no baseline (clinical) variables to be collected at the onset? Or values of baseline variables have to meet certain threshold before admission to the trial? If so, eligibility can be confirmed on site during the baseline interview. 

As the first contact with potential participants is usually made by telephone to conduct the study information, possible exclusion criteria, such as a primary caesarean section, are already identified (see Section “Participants”). The exclusion criteria are then comprehensively checked in the baseline survey and women are excluded on the basis of these.

L180: What is the interpretation of an effect size of 0.75 in the context of the trial. 

Based on previous comparable studies, the moderate to high effect size of 0.75 was set. This assumption would mean that the BreLax intervention could have a substantial impact on self-efficacy (Leutenegger et al., 2022).

L183: The 20% attrition rate comes from nowhere without justifications. This results in a sample size of of 35 subjects to be recruited for each arm. The hospital has 1800 births and there was a brief discussion of contamination. So why not perform the trial staggered over time to minimize contamination?

The study is part of a PhD project and is therefore limited in terms of both time and funding. For this reason, although a phased implementation was discussed, it was not considered feasible for the reasons mentioned. The 20% attrition rate is due to the fact that the women in the clinic are cared for relatively closely by the gynaecological specialists and midwives, which is why a loss through follow-up is rather unlikely. In addition, a 20% attrition rate was recommended in the statistical consultation due to the academic degree of the project manager.

L189: Why have the midwife use envelopes to randomize subjects into group membership. It is the same reason why you don’t want the same statistician to randomize the trial and perform the analysis.

Yes, that is correct. The study midwife has allocated all classes taking place in the coming year with the help of envelopes during the study planning phase.

L334 ‘the possibility of multiple imputation’ will be examined. This is very vague. Please elaborate. Identify situations if and when you will use multiple imputation.

In the proposed study, we will explore the possibility of multiple imputation when we encounter missing data. The decision to use multiple imputation will depend on the amount and type of missing data and the requirements of our analysis.

L344 You also reference doing the analyses using a serious of linear mixed effects models. What models do you have in mind? Please provide more details and display the regression models and show what are the random and fixed effects for added clarity

During the planning of the analysis, the random intercept model and the random intercept and slope model were identified. The random intercept model would allow the baseline self-efficacy data to vary in the direction of birth. The random intercept and slope model if the effect of the BreLax intervention is expected to vary between women in the study. The randomised slope would allow for each woman to have her own unique response to the intervention over time. For the following reasons, this model was assumed in the protocol.

Further comments: Throughout, the methodology seems to assume the outcome is normally distributed. Where are the checks and what is the proposed methodology if they are not normally distributed?

Thank you for pointing this out. These were addressed individually in the respective points (data analysis) and a specific explanation was provided in the revised manuscript in the section on data analysis (Line 326-354).

Reviewer2# Further comments:

Thank you very much for your comments. The abstract has been revised according to the word recommendation. The citation style has been changed accordingly and the language corrections have been made. 

Participating women register themselves for their chosen antenatal education classes. During the planning phase, all advertised courses for the coming year were randomised (intervention yes / no), which is why the women do not know which course they are attending. The information section on breathing and relaxation exercises is included in both classes, but not the exercise sequences or the online guide. This point could potentially be critical when the women discuss possible exercises within the course. In order to identify this bias, the last questionnaire collects information on this, such as "Have you had contact with women from other courses?", "Did you receive any further information?".

---

## [Editor Report · Decision Letter 1]

25 Jul 2024

Study protocol of a breathing and relaxation intervention included in antenatal education: a randomised controlled trial (BreLax study)

PONE-D-24-06197R1

Dear Dr. Leutenegger,

We’re pleased to inform you that your manuscript has been judged scientifically suitable for publication and will be formally accepted for publication once it meets all outstanding technical requirements.

Kind regards,

David Chibuike Ikwuka, Ph.D.

Academic Editor

PLOS ONE
---

## [Editor Report · Acceptance letter]

6 Aug 2024

PONE-D-24-06197R1 

PLOS ONE

Dear Dr. Leutenegger, 

I'm pleased to inform you that your manuscript has been deemed suitable for publication in PLOS ONE. Congratulations! Your manuscript is now being handed over to our production team.

Kind regards, 

on behalf of

Dr David Chibuike Ikwuka 

Academic Editor

PLOS ONE